# Building Generalizable Agents with a Realistic and Rich 3D Environment

**Yi Wu**
UC Berkeley
`jxwuyi@gmail.com`

**Yuxin Wu & Georgia Gkioxari & Yuandong Tian**
Facebook AI Research
`{yuxinwu,gkioxari,yuandong}@fb.com`

## Abstract

Teaching an agent to navigate in an unseen 3D environment is a challenging task, even in the event of simulated environments. To generalize to unseen environments, an agent needs to be robust to low-level variations (e.g. color, texture, object changes), and also high-level variations (e.g. layout changes of the environment). To improve overall generalization, all types of variations in the environment have to be taken under consideration via different level of data augmentation steps. To this end, we propose *House3D*, a rich, extensible and efficient environment that contains 45,622 human-designed 3D scenes of visually realistic houses, ranging from single-room studios to multi-storied houses, equipped with a diverse set of fully labeled 3D objects, textures and scene layouts, based on the SUNCG dataset (Song et al., 2017). The diversity in House3D opens the door towards scene-level augmentation, while the label-rich nature of House3D enables us to inject pixel- & task-level augmentations such as domain randomization (Tobin et al., 2017) and multi-task training. Using a subset of houses in House3D, we show that reinforcement learning agents trained with an enhancement of different levels of augmentations perform much better in unseen environments than our baselines with raw RGB input by over $8\%$ in terms of navigation success rate. House3D is publicly available at `http://github.com/facebookresearch/House3D`.

## 1 Introduction

Recently, deep reinforcement learning has shown its strength on multiple games, such as Atari (Mnih et al., 2015) and Go (Silver et al., 2016), vastly overpowering human performance. Via the various reinforcement learning frameworks, different aspects of intelligence can be learned, including 3D understanding (DeepMind Lab (Beattie et al., 2016) and Malmo (Johnson et al., 2016)), real-time strategy decision (TorchCraft (Synnaeve et al., 2016) and ELF (Tian et al., 2017)), fast reaction (Atari (Bellemare et al., 2013)), long-term planning (Go, Chess), language and communications (ParlAI (Miller et al., 2017) and (Das et al., 2017b)).

A prominent issue in reinforcement learning is *generalizability*. Commonly, agents trained on a specific environment and for a specific task become highly specialized and fail to perform well on new environments. In the past, there have been efforts to address this issue. In particular, pixel-level variations are applied to the observation signals in order to increase the agent's robustness to unseen environments (Beattie et al., 2016; Higgins et al., 2017; Tobin et al., 2017). Parametrized environments with varying levels of difficulty are used to yield scene variations but with similar visual observations (Pathak et al., 2017). Transfer learning is applied to similar tasks but with different rewards (Finn et al., 2017b).

Nevertheless, the aforementioned techniques study the problem in simplified environments which lack the diversity, richness and perception challenges of the real world. To this end, we propose a substantially more diverse environment, *House3D*, to train and test our agents. House3D is a

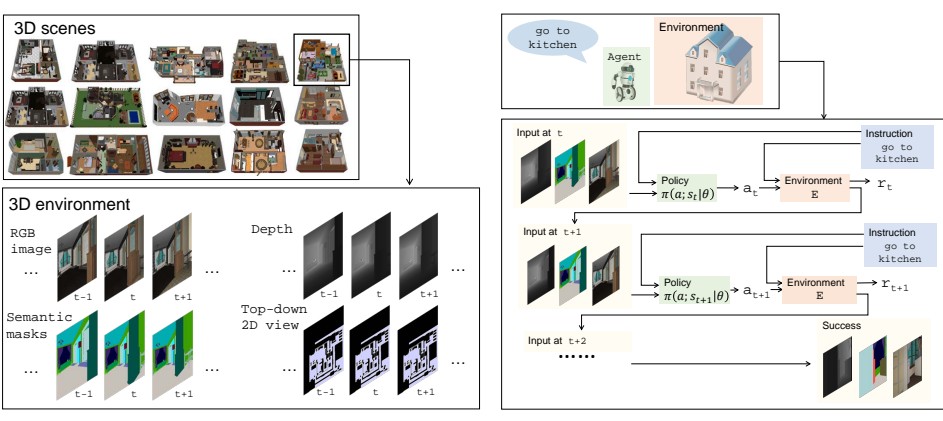

(a) House3D environment                    (b) RoomNav task

Figure 1: An overview of House3D environment and RoomNav task. **(a)** We build an efficient and interactive environment upon the SUNCG dataset (Song et al., 2017) that contains 45K diverse indoor scenes, ranging from studios to two-storied houses with swimming pools and fitness rooms. All 3D objects are fully labeled into over 80 categories. Observations of agents in the environment have multiple modalities, including RGB images, Depth, Segmentation masks (from object category), top-down 2D view, etc. **(b)** We focus on the task of targeted navigation. Given a high-level description of a room concept, the agent explores the environment to reach the target room.

virtual 3D environment consisting of thousands of indoor scenes equipped with a diverse set of scene types, layouts and objects. An overview of House3D is shown in Figure 1a. House3D leverages the SUNCG dataset (Song et al., 2017) which contains 45K *human-designed* real-world 3D house models, ranging from single studios to houses with gardens, in which objects are fully labeled with categories. We convert the SUNCG dataset to an *environment*, House3D, which is efficient and extensible for various tasks. In House3D, an agent can freely explore the space while perceiving a large number of objects under various visual appearances.

Based on House3D, we design a task called *RoomNav*: an agent starts at a random location in a house and is asked to navigate to a destination specified by a high-level semantic concept (*e.g. kitchen*), following simple rules (*e.g.* no object penetration), as shown in Figure 1b. We use gated-CNN and gated-LSTM policies trained with standard deep reinforcement learning methods, *i.e.* A3C (Mnih et al., 2016) and DDPG (Lillicrap et al., 2015), and report success rate on unseen environments over 5 concepts. We show that in order to achieve strong generalization capability, **all-levels of augmentations are needed**: pixel-level augmentation by domain randomization (Tobin et al., 2017) enhances the agent's robustness to color variations; object-level augmentation forces the agent to learn multiple concepts (20 in number) simultaneously, and scene-level augmentation, where a diverse set of environments is used, enforce generalizability across diverse scenes, mitigating overfitting to particular scenes. Our final gated-LSTM agent achieves a success rate of $35.8\%$ on 50 unseen environments, 10% better than the baseline method (25.7%).

The remaining of the paper is structured as follows. Section 2 summarizes relevant work. Section 3 describes our environment, House3D, in detail and section 4 describes the task, RoomNav. Section 5 describes our gated models and the applied algorithms to tackle RoomNav. Finally, experimental results are shown in Section 6.

## 2 RELATED WORK

**Environments**: Table 1 shows the comparison between House3D and most relevant prior works. There are other simulated environments which focus on different domains, such as OpenAI Gym (Brockman et al., 2016), ParlAI (Miller et al., 2017) for language communication as well as some strategic game environments (Synnaeve et al., 2016; Tian et al., 2017; Vinyals et al., 2017), etc. Most of these environments are pertinent to one particular aspect of intelligence, such as dialogue or a single type of game, which makes it hard to facilitate the study of more comprehensive

| Environment | 3D | Realistic | Large-scale | Fast | Customizable |
|---|:---:|:---:|:---:|:---:|:---:|
| Atari (Bellemare et al., 2013) | | | | ● | |
| OpenAI Universe (Shi et al., 2017) | | ● | ● | | ● |
| Malmo (Johnson et al., 2016) | ● | | ● | ● | ● |
| DeepMind Lab (Beattie et al., 2016) | ● | | | ● | ● |
| VizDoom (Kempka et al., 2016) | ● | | | ● | ● |
| AI2-THOR (Zhu et al., 2017) | ● | ● | | ● | |
| Stanford2D-3D (Armeni et al., 2016) | ● | ● | | ● | |
| Matterport3D (Chang et al., 2017) | ● | ● | ● | ● | |
| House3D | ● | ● | ● | ● | ● |

Table 1: A summary of popular environments. The attributes include **3D**: 3D nature of the rendered objects, **Realistic**: resemblance to the real-world, **Large-scale**: a large set of environments, **Fast**: fast rendering speed and **Customizable**: flexibility to be customized to other applications.

problems. On the contrary, we focus on building a platform that intersects with multiple research directions, such as object and scene understanding, 3D navigation, embodied question answering (Das et al., 2017a), while allowing users to customize the level of complexity to their needs.

We build on SUNCG (Song et al., 2017), a dataset that consists of thousands of diverse synthetic indoor scenes equipped with a variety of objects and layouts. Its visual diversity and rich content opens the path to the study of semantic generalization for reinforcement learning agents. Our platform decouples high-performance rendering from data I/O, and thus can use other publicly available 3D scene datasets as well. This includes Al2-THOR (Zhu et al., 2017), SceneNet RGB-D (McCormac et al., 2017), Stanford 3D (Armeni et al., 2016), Matterport 3D (Chang et al., 2017) and so on.

Concurrent works (Brodeur et al., 2017; Savva et al., 2017) also introduce similar platforms as House3D, indicating the interest for large-scale interactive and realistic 3D environments.

**3D Navigation**: There has been a prominent line of work on the task of navigation in real 3D scenes (Leonard & Durrant-Whyte, 1992). Classical approaches decompose the task into two subtasks by building a 3D map of the scene using SLAM and then planning in this map (Fox et al., 2005). More recently, end-to-end learning methods were introduced to predict robotic actions from raw pixel data (Levine et al., 2016). Some of the most recent works on navigation show the effectiveness of end-to-end learning. Gupta et al. (2017) learn to navigate via mapping and planning using shortest path supervision. Sadeghi & Levine (2017) teach an agent to fly using solely simulated data and deploy it in the real world. Dhiraj et al. (2017) collect a dataset of drones crashing into objects and train self-supervised agents on this data to avoid obstacles.

A number of recent works also use deep reinforcement learning for navigation in simulated 3D scenes. Mirowski et al. (2016); Jaderberg et al. (2016) improve an agent's navigation ability in mazes by introducing auxiliary tasks. Parisotto & Salakhutdinov (2017) propose a new architecture which stores information of the environment on a 2D map. Karl Moritz Hermann & PhilBlunsom (2017) focus on the task of language grounding by navigating simple 3D scenes. However, these works only evaluate the agent's generalization ability on pixel-level variations or small mazes. We argue that a much richer environment is crucial for evaluating semantic-level generalization.

**Gated Modules**: In our work, we focus on the task of RoomNav, where the goal is communicated to the agent as a high-level instruction selected from a set of predefined concepts. To modulate the behavior of the agent in RoomNav, we encode the instruction as an embedding vector which gates the visual signal. The idea of gated attention has been used in the past for language grounding (Chaplot et al., 2017), and transfer learning by language grounding (Narasimhan et al., 2017). Similar to those works, we use concept grounding as an attention mechanism. We believe that our gated reinforcement learning models serve as a strong baseline for the task of semantic based navigation in House3D. Furthermore, our empirical results allow us to draw conclusions on the models' efficacy when training agents in a large-scale, diverse dataset with an emphasis on generalization.

**Generalization**: There is a recent trend in reinforcement learning focusing on the problem of generalization, ranging from learning to plan (Tamar et al., 2016), meta-learning (Duan et al., 2016; Finn et al., 2017a) to zero-shot learning (Andreas et al., 2016; Oh et al., 2017; Higgins et al., 2017).

However, these works either focus on over-simplified tasks or test on environments which are only slightly varied from the training ones. In contrast, we use a more diverse set of environments, each containing visually and structurally different observations, and show that the agent can work well in unseen scenes.

In this work, we show improved generalization performance in complex 3D scenes when using depth and segmentation masks on top of the raw visual input. This observation is similar to other works which use a diverse set of input modalities (Mirowski et al., 2016; Tai & Liu, 2016). Our result suggests that it can be possible to decouple real-world robotics from recognition via a vision API provided by an object detection or semantic segmentation system trained on the targeted real scenes. This opens the door towards bridging the gap between simulated environment and real-world (Tobin et al., 2017; Rusu et al., 2016; Christiano et al., 2016).

## 3 HOUSE3D: AN EXTENSIBLE ENVIRONMENT OF 45K 3D HOUSES

We propose House3D, an environment which closely resembles the real world and is rich in content and structure. An overview of House3D is shown in Figure 1a. House3D is developed to provide an efficient and flexible environment of thousands of indoor scenes and facilitates a variety of tasks, *e.g.* navigation, visual understanding, language grounding, concept learning etc. The environment along with a python API for easy use is available at `http://github.com/facebookresearch/House3D`.

### 3.1 DATASET

The 3D scenes in House3D are sourced from the SUNCG dataset (Song et al., 2017), which consists of 45,622 human-designed 3D scenes ranging from single-room studios to multi-floor houses. The SUNCG dataset was designed to encourage research on large-scale 3D object recognition problems and thus carries a variety of objects, scene layouts and structures. On average, there are 8.9 rooms and 1.3 floors per scene There is a diverse set of room and object types in each scene. In total, there are over 20 different room types, such as bedroom, living room, kitchen, bathroom etc., with over 80 different object categories. In total, the SUNCG dataset contains 404,508 different rooms and 5,697,217 object instances drawn from 2644 unique object meshes.

### 3.2 ANNOTATIONS

Each scene in SUNCG is fully annotated with 3D coordinates and its room and object types (e.g. bedroom, shoe cabinet, etc). This allows for a detailed mapping from each 3D location to an object instance (or None at free space) and the room type.

At every time step an agent has access to the following signals: a) the visual RGB signal of its current first person view, b) semantic/instance segmentation masks for all the objects visible in its current view, and c) depth information. For different tasks, these signals might serve for different purposes, e.g., as a feature plane or an auxiliary target. Based on the existing annotations, House3D offers more information, e.g., top-down 2D occupancy maps, connectivity analysis and shortest paths between two points.

### 3.3 RENDERER

To build a realistic 3D environment, we develop a renderer for the SUNCG scenes. The renderer is based on OpenGL, it can run on both Linux and MacOS, and provides RGB images, semantic segmentation masks, instance segmentation masks and depth maps.

As highlighted above, the environment needs to be *efficient* in order to be used for large-scale reinforcement learning. On a NVIDIA Tesla M40 GPU, our implementation can render $120 \times 90$-sized frames at over 600 fps, while multiple renderers can run in parallel on one or more GPUs. When rendering multiple houses simultaneously, one M40 GPU can be fully utilized to render at a total of 1800 fps. The default simple physics adds a small overhead to the rendering. The high throughput of our implementation enables efficient learning for a variety of interactive tasks, such as on-policy reinforcement learning.

### 3.4 INTERACTION

In House3D, an agent can live in any location within a 3D scene, as long as it does not collide with object instances (including walls) within a small range, *i.e.* robot's radius. Doors, gates and arches are considered passage ways, meaning that an agent can walk through those structures freely. These default design choices add negligible run-time overhead. Note that more complex interaction rules can be incorporated (*e.g.* manipulation) within House3D using our flexible API, which we leave for future work.

## 4 ROOMNAV: A BENCHMARK TASK FOR CONCEPT-DRIVEN NAVIGATION

Consider the task of concept-driven navigation as shown in Figure 1b. A human may give a high level instruction to the robot, for example, "Go to the *kitchen*", so that one can later ask the robot to turn on the oven. The robot needs to behave appropriately conditioned on the house it is located in and the goal, *e.g.* the semantic concept "kitchen". In addition, we want the agent to *generalize*, *i.e.* to perform well in unseen environments, that is new houses with different layouts and furniture locations.

To study the aforementioned abilities of an agent, we develop a benchmark task, *Concept-Driven Navigation* (RoomNav), based on House3D. We define the goal to be of the form "go to X", where X denotes a pre-defined room type or object type, which is a semantic concept that an agent needs to interpret from a variety of scenes of distinct visual appearances. To ensure fast experimentation cycles, we perform experiments on a subset of House3D. We manually select 270 houses suitable for a navigation task and split them into a small set (20 houses), a large set (200 houses) and a test set (50 houses), where the test set is used to evaluate the generalization of the trained agents.

**Task Formulation:** Suppose we have a set of episodic environments $\mathcal{E} = \{E_1, .., E_n\}$ and a set of semantic concepts $\mathcal{I} = \{I_1, .., I_m\}$. During each episode, the agent is interacting with one environment $E \in \mathcal{E}$ and is given a concept $I \in \mathcal{I}$. In the beginning of an episode, the agent is randomly placed somewhere in $E$. At each time step $t$, the agent receives a visual signal $X_t$ from $E$ via its first person view sensor. Let $s_t = \{X_1, .., X_t, I\}$ denote the state of the agent at time $t$. The agent needs to propose an action $a_t$ to navigate and rotate its sensor given $s_t$. The environment returns a reward signal $r_t$ and terminates when the agent succeeds in finding the destination, or reaches a maximum number of steps.

The objective of this task is to learn an optimal policy $\pi(a_t|s_t, I)$ that leads to the target defined by $I$. We train the agent on a set $\mathcal{E}_{\text{train}}$. We evaluate the policy on a disjoint set of environments $\mathcal{E}_{\text{test}}$ ( $\mathcal{E}_{\text{test}} \cap \mathcal{E}_{\text{train}} = \emptyset$). For more details see the Appendix.

**Environment Statistics:** The selected 270 houses are manually verified for navigation; they are well connected, contain desired concepts, and are large enough for exploration. We split them into 3 disjoint sets, denoted by $\mathcal{E}_{small}$, $\mathcal{E}_{large}$ and $\mathcal{E}_{test}$ respectively. For the semantic concepts, we select the five most common room types: kitchen, living room, dining room, bedroom and bathroom. Note that this set can be extended to include objects or even subareas within rooms.

**Observations:** We utilize three different kinds of visual input signals for $X_t$, including (1) raw pixel values; (2) semantic segmentation mask of the pixel input; and (3) depth information, and experiment with different combinations of them. We encode each concept $I$ as a one-hot vector representation.

**Action Space:** Similar to existing navigation works, we define a fixed set of actions, here 12 in number including different scales of rotations and movements. Due to the complexity of the indoor scenes, we also explore a continuous action space similar to (Lowe et al., 2017), which in effect allows the agent to move with different velocities. For more details see the Appendix. In all cases, if the agent hits an obstacle it remains still.

**Success Measure and Reward Function:** To declare *success*, we want to ensure that the agent *identifies* the target room by its unique properties (*e.g.* presence of appropriate objects in the room such as pan and knives for kitchen and bed for bedroom) instead of merely reaching there by luck. An episode is considered successful if both of the following two criteria are satisfied: (1) the agent is located inside the target room; (2) the agent consecutively *sees* a designated object category

associated with that target room type for at least 2 time steps. We assume that an agent *sees* an object if there are at least 4% of pixels in $X_t$ belonging to that object.

For the reward function, ideally two signals suffice to reflect the task requirement: (1) a collision penalty when hitting obstacles; and (2) a success reward when completing the task. However, these basic signals make it too difficult for an RL agent to learn, as the positive reward is too sparse. To provide additional supervision during training, we resort to an informative reward shaping: we compute the approximate shortest distance from the target room to each location in the house and adopt the difference of shortest distances between the agent's movement as an additional reward signal. Note that our ultimate goal is to learn a policy that could *generalize* to unseen houses. Our strong reward shaping supervises the agent at training and is *not available* to the agent at test time. We empirically observe that stronger reward shaping leads to better performances on both training and testing.

## 5 GATED-ATTENTION NETWORKS FOR MULTI-TARGET LEARNING

The RoomNav task can be considered as a multi-target learning problem: the policy needs to condition on both the input $s_t$ and the target concept $I$. For policy representations which incorporate the target $I$, we propose two baseline models with a gated-attention architecture, similar to Dhingra et al. (2016) and Chaplot et al. (2017): a gated-CNN network for continuous actions and a gated-LSTM network for discrete actions. We train the gated-CNN policy using the deep deterministic policy gradient (DDPG) (Lillicrap et al., 2015), while the gated-LSTM policy is trained using the asynchronous advantage actor-critic algorithm (A3C) (Mnih et al., 2016).

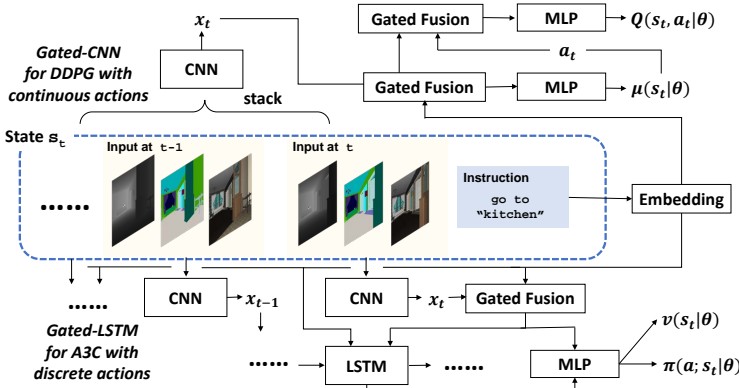

Figure 2: Overview of our proposed models. Bottom part demonstrates the gated-LSTM model for discrete action while the top part shows the gated-CNN model for continuous action. The "Gated Fusion" module denotes the gated-attention architecture.

### 5.1 DDPG WITH GATED-CNN POLICY

#### 5.1.1 DEEP DETERMINISTIC POLICY GRADIENT

Suppose we have a deterministic policy $\mu(s_t|\theta)$ (actor) and the Q-function $Q(s_t, a|\theta)$ (critic) both parametrized by $\theta$. DDPG optimizes the policy $\mu(s_t|\theta)$ by maximizing $L_\mu(\theta) = \mathbb{E}_{s_t}[Q(s_t, \mu(s_t|\theta)|\theta)]$, and updates the Q-function by minimizing $L_Q(\theta) = \mathbb{E}\left[(Q(s_t, a_t|\theta) - \gamma Q(s_{t+1}, \mu(s_{t+1}|\theta)|\theta) - r_t)^2\right]$.

Here, we use a shared network for both actor and critic with the final loss function $L_{\text{DDPG}}(\theta) = -L_\mu(\theta) + \alpha_{\text{DDPG}} L_Q(\theta)$, where $\alpha_{\text{DDPG}}$ is a constant balancing the two objectives.

#### 5.1.2 GATED-CNN FOR CONTINUOUS POLICY

**State Encoding:** Given state $s_t$, we first stack the most recent $k$ frames $X = [X_t, X_{t-1}, \ldots, X_{t-k+1}]$ channel-wise and apply a convolutional neural network to derive an im-

age representation $x = f_{\text{cnn}}(X|\theta) \in \mathbb{R}^{d_X}$. We convert the target $I$ into an embedding vector $y = f_{\text{embed}}(I|\theta) \in \mathbb{R}^{d_I}$. Subsequently, we apply a fusion module $M(x, y|\theta)$ to derive the final encoding $h_s = M(x, y|\theta)$.

**Gated-Attention for Feature Fusion:** For the fusion module $M(x, y|\theta)$, the straightforward version is concatenation, namely $M_{\text{cat}}(x, y|\cdot) = [x, y]$. In our case, $x$ is always a high-dimensional feature vector (i.e., image feature) while $y$ is a simple low-dimensional conditioning vector (e.g., instruction). Thus, simple concatenation may result in optimization difficulties. For this reason, we propose to use a gated-attention mechanism. Suppose $x \in \mathbb{R}^{d_x}$ and $y \in \mathbb{R}^{d_y}$ where $d_y < d_x$. First, we transform $y$ to $y' \in \mathbb{R}^{d_x}$ via an MLP, namely $y' = f_{\text{mlp}}(y|\theta)$, and then perform a Hadamard (pointwise) product between $x$ and sigmoid($y'$), which leads to our final gated fusion module $M(x, y|\theta) = x \odot \text{sigmoid}(f_{\text{mlp}}(y|\theta))$. This gated fusion module could also be interpreted as an attention mechanism over the feature vector which could help better shape the feature representation.

**Policy Representation:** For the policy, we apply a MLP layer on the state representation $h_s$, followed by a softmax operator (for bounded velocity) to produce the continuous action. Moreover, in order to produce a stochastic policy for both better exploration and higher robustness, we apply the Gumbel-Softmax trick (Jang et al., 2016), resulting in the final policy $\mu(s_t|\theta) = \text{Gumbel-Softmax}(f_{\text{mlp}}(h_s|\theta))$. Note that since we add randomness to $\mu(s_t|\theta)$, our DDPG formulation can also be interpreted as the SVG(0) algorithm (Heess et al., 2015).

**Q-function:** The Q-function $Q(s, a)$ conditions on both state $s$ and action $a$. We again apply a gated fusion module to the feature vector $x$ and the action vector $a$ to derive a hidden representation $h_Q = M(x, a|\theta)$. We eventually apply another MLP to $h_Q$ to produce the final value $Q(s, a)$.

A model demonstration is shown in the top part of Fig. 2, where each block has its own parameters.

## 5.2 A3C WITH GATED-LSTM POLICY

### 5.2.1 ASYNCHRONOUS ADVANTAGE ACTOR-CRITIC

Suppose we have a discrete policy $\pi(a; s|\theta)$ and a value function $v(s|\theta)$. A3C optimizes the policy by minimizing the loss function $L_{\text{pg}}(\theta) = -\mathbb{E}_{s_t, a_t, r_t} \left[ \sum_{t=1}^{T} (R_t - v(s_t)) \log \pi(a_t; s_t|\theta) \right]$, where $R_t$ is the discounted accumulative reward defined by $R_t = \sum_{i=0}^{T-t} \gamma^i r_{t+i} + v(s_{T+1})$. The value function is updated by minimizing the loss $L_v(\theta) = \mathbb{E}_{s_t, r_t}[(R_t - v(s_t))^2]$.

Finally the overall loss function for A3C is $L_{\text{A3C}}(\theta) = L_{\text{pg}}(\theta) + \alpha_{\text{A3C}} L_v(\theta)$ where $\alpha_{\text{A3C}}$ is a constant coefficient.

### 5.2.2 GATED-LSTM NETWORK FOR DISCRETE POLICY

**State Encoding:** Given state $s_t$, we first apply a CNN module to extract image feature $x_t$ for each input frame $X_t$. For the target, we apply a gated fusion module to derive a state representation $h_t = M(x_t, I|\theta)$ at each time step $t$. Then, we concatenate $h_t$ with the target $I$ and the result is fed into the LSTM module (Hochreiter & Schmidhuber, 1997) to obtain a sequence of LSTM outputs $\{o_t\}_t$, so that the LSTM module has direct access to the target other than the attended visual feature.

**Policy and Value Function:** For each time step $t$, we concatenate the state vector $h_t$ with the output of the LSTM $o_t$ to obtain a joint hidden vector $h_{\text{joint}} = [h_t, o_t]$. Then we apply two MLPs to $h_{\text{joint}}$ to obtain the policy distribution $\pi(a; s_t|\theta)$ as well as the value function $v(s_t|\theta)$.

A visualization of the model is in the bottom part of Fig. 2. The parameters of CNN modules are shared across time.

## 6 EXPERIMENTAL RESULTS

We report experimental results for our models on the task of RoomNav. We first compare models with discrete and continuous action spaces with different input modalities. Then we explain our

observations and show that techniques targeting different levels of augmentation improve the success rate of navigation in the test set. Moreover, these techniques are complementary to each other.

**Setup.** We train our baseline models on multiple experimental settings. We use two training datasets. The small set $\mathcal{E}_{small}$ contains 20 houses and the large set $\mathcal{E}_{large}$ contains 200 houses. A held-out dataset $\mathcal{E}_{test}$ is used for test, which contains 50 houses.

We mainly focus on success rate on the test set, i.e, how the agent generalizes. For reference, we also report the training performance. The agent fails if it failed to find the concept within 100 steps[1]. All success rate evaluations use a fixed random seed for a fair comparison. For each model, we run 2000 evaluation episodes on $\mathcal{E}_{small}$ and $\mathcal{E}_{test}$, and 5000 evaluation episodes on $\mathcal{E}_{large}$ to measure overall success rates.

We use `gated-CNN` and `gated-LSTM` to denote the networks with gated-attention, and `concat-CNN` and `concat-LSTM` for models with simple concatenation. We also experiment with different visual signals to the agents, including RGB image (RGB Only), RGB image with depth information (RGB+Depth) and semantics mask with depth information (Mask+Depth). The input image resolution is $120 \times 90$ to preserve image details.

During each simulated episode, we randomly select a house from the environment set and randomly pick an applicable target from the house to instruct the agent. During training, we add an entropy bonus term for both models[2] in addition to the original loss function. For evaluation, we keep the final model for DDPG due to its stable learning curve, while for A3C, we take the model with the highest training success rate. We use Pytorch (Paszke et al., 2017) and Adam (Kingma & Ba, 2014). See Appendix for more experiment details.

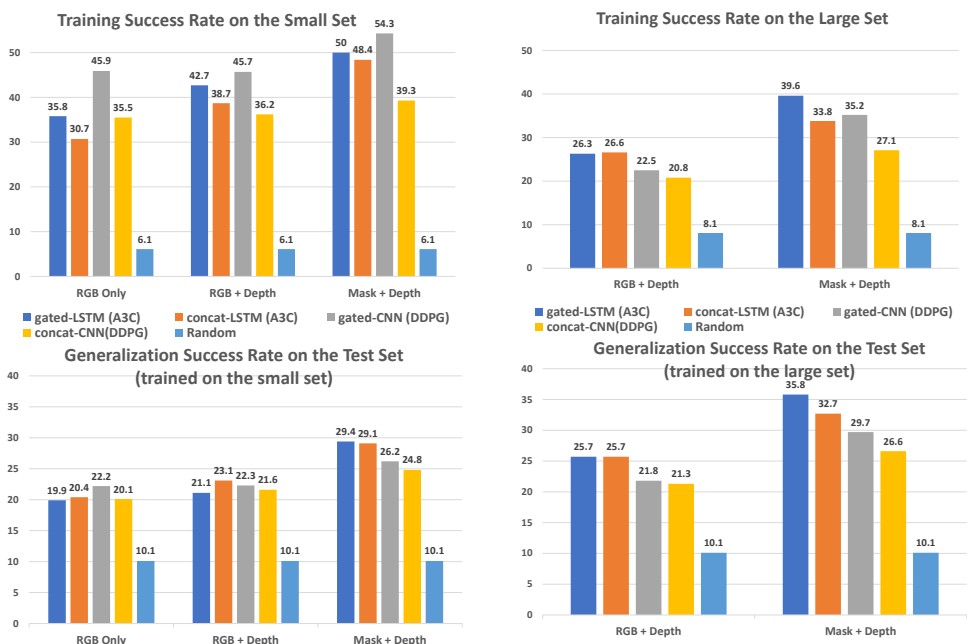

(a) Training (top) and test (bottom) results on small set

(b) Training (top) and test (bottom) results on large set

Figure 3: **Overall** performance of various models trained on (a) $\mathcal{E}_{small}$ (20 houses) with different input signals: RGB Only, RGB+Depth and Mask+Depth; (b) $\mathcal{E}_{large}$ (200 houses) with input signals: RGB+Depth and Mask+Depth. In each group, the bars from left to right correspond to gated-LSTM, concat-LSTM, gated-CNN, concat-CNN and random policy respectively.

---

[1]This is enough for success evaluation. The average number of steps for success runs in every setting is less than 45, which is much smaller than 100. Refer to appendix B.4 for details.

[2]For DDPG, we simply use the entropy of the softmax output.

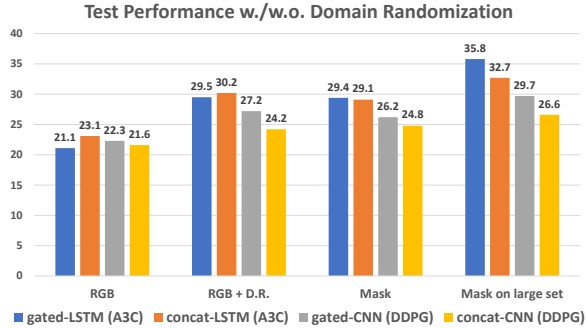

Figure 4: **Pixel-level Augmentation:** Test performances of various models trained with different input signals, including RGB+Depth on $\mathcal{E}_{\text{small}}$, RGB with Domain Randomization on $\mathcal{E}_{\text{small}}$, Mask+Depth on $\mathcal{E}_{\text{small}}$, Mask+Depth on $\mathcal{E}_{\text{large}}$. In each group, the bars represent gated-LSTM, concat-LSTM, gated-CNN and concat-CNN from left to right.

### 6.1 BASELINES: MODELS WITH RGB SIGNALS ON $\mathcal{E}_{\text{SMALL}}$

As shown in the bottom part of Fig. 3a, on $\mathcal{E}_{\text{small}}$, the test success rate for models trained on RGB features is unsatisfactory. We observe obvious overfitting behavior: the test performance is drastically worse than training. In particular, the gated-LSTM models achieve even lower success rate than concat-LSTM models, despite the fact that they have much better training performance. In this case, the learning algorithm picks up spurious color patterns in the environments as the guidance towards the goal, which is inapplicable to unseen environments.

In both training and test, we find that depth information improves the performance thus we use it in the following experiments and omit *Depth* for conciseness.

### 6.2 TECHNIQUES FOR DIFFERENT LEVELS OF AUGMENTATION

Augmentation is a standard technique to improve generalization. However, for complicated tasks, augmentation needs to be taken care at different levels. In this section, we categorize augmentation techniques into 3 levels: (1) pixel-level augmentation: changing the colors and textures; (2) task-level augmentation: joint learning for multiple tasks; (3) scene-level augmentation: training on more environments. We analyze the generalization performance with all techniques and conclude that these techniques are complementary and that the best test performance is obtained by combining these techniques together.

**Pixel-level Augmentation:** We use domain randomization (Tobin et al., 2017), by reassigning each object in the scene a random color but keeping the textures. This breaks the spurious color correlations and pushes the agent to learn a better representation.

We explore domain randomization by generating an additional 180 houses with random object coloring from $\mathcal{E}_{\text{small}}$, which leads to a total of 200 houses. We evaluate the test success rate of various models under different training settings, e.g., RGB, RGB with domain randomization (D.R.) or mask signal. The results are shown in Fig. 4. Interestingly, we noticed that domain randomization yields very similar performance as mask signal on $\mathcal{E}_{\text{small}}$.

One shortcoming of domain randomization is that it requires substantially more training samples and thus suffers from high sample complexity. Thanks to the rich labels in House3D, we instead could use segmentation mask as an input feature plane, which encodes semantic information and is independent of the object color. This helps train generalizable agent with much fewer training samples. On the other hand, an agent trained with domain randomization can operate with RGB input only, without segmentation mask output from a vision subsystem. In the current context, we simply assume adopting segmentation mask input as the technique for pixel-level augmentation.

**Task-level Augmentation:** We explore task-level augmentation by adding related auxiliary targets during training (Fig. 5). Specifically, in addition to the 5 room types as auxiliary targets, we selected

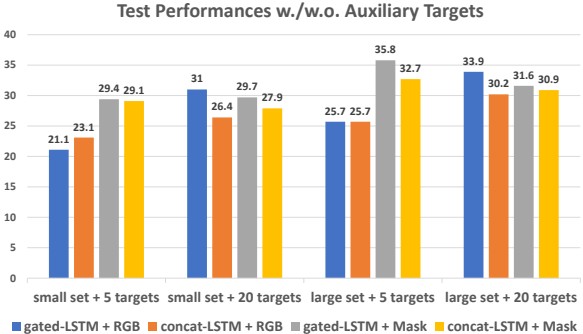

Figure 5: **Task-Level Augmentation:** Test performances of LSTM models trained with and without auxiliary targets on both $\mathcal{E}_{\text{small}}$ and $\mathcal{E}_{\text{large}}$. In each group, the bars represent gated-LSTM + RGB, concat-LSTM + RGB, gated-LSTM + Mask and concat-LSTM + Mask from left to right.

15 object concepts (e.g., chair, table, cabinet, etc. See a full list of object concepts in appendix.). We train A3C agents with different input signals on $\mathcal{E}_{\text{small}}$ and evaluate their test performances.

We found that auxiliary targets significantly reduce overfitting and increases the generalizability of models with RGB inputs. Because of this effect, gated attention model, which has high model capacity, becomes much more effective on RGB signal when trained with more targets. On the other hand, with mask input, the agent does not need to learn to differentiate the objects, therefore auxiliary targets do not help that much for more complicated models like gated attention models.

**Scene-level Augmentation:** We could further boost the generalization performance by augmenting the training set with more diverse set of houses, i.e, $\mathcal{E}_{\text{large}}$ that contain 200 different houses. This is also a benefit from House3D.

For visual signals, we focus on feature combinations like "RGB + Depth" and "Mask + Depth". Note that for training efficiency, segmentation mask is a surrogate feature to approximate "RGB + domain randomization" as it shows similar results in the small set. Both train and test results are summarized in Fig. 3b.

On a semantically diverse dataset $\mathcal{E}_{\text{large}}$, the overfitting issue is largely resolved. We see drops in the training performance and improve on the generalization. After training on a large number of environments, every model now has a much smaller gap between its training and test performance. This is in particularly true for the models using RGB signal, which suffers from overfitting issues on $\mathcal{E}_{\text{small}}$. Notably, on large dataset, LSTM models generally perform better than CNN models due to its high model capacity.

In addition, similar behavior was also observed during our experiments with techniques for pixel-level augmentation (Fig. 4) and task-level augmentation (Fig. 5). In all the experiments, all the models consistently achieves better generalization performances when trained on $\mathcal{E}_{\text{large}}$, which again emphasizes the benefits of House3D.

The overall best success rate is achieved by gated-attention architecture with semantic signals. It is better than both RGB channels by over 8% and the counterpart trained on $\mathcal{E}_{\text{small}}$ in terms of generalization metric. This means that pixel-level augmentation (e.g., domain randomization and/or segmentation mask) and scene-level augmentation (e.g., using diverse dataset) can improve the performance. Moreover, their effects are complementary.

A diverse environment like $\mathcal{E}_{\text{large}}$ also enables the model of larger capacity to work better. For example, LSTMs considerably outperform the simpler reactive models, i.e., CNNs with recent 5 frames as state input. We believe this is due to the larger scale and the high complexity of the training set, which makes it almost impossible for an agent to "remember" the optimal actions for every scenario. Instead, an agent needs to develop high-level abstractions (e.g., high-level exploration strategy, memory, etc). These are helpful induction biases that could lead to a more generalizable model.

Lastly, we also analyze the detailed success rate with respect to each target room in appendix.

# 7 CONCLUSION

In this paper, we propose a new environment, House3D, which contains 45K houses with a diverse set of objects and natural layouts resembling the real-world.

In House3D, we teach an agent to accomplish semantic goals. We define RoomNav, in which an agent needs to understand a given semantic concept, interpret the comprehensive visual signal, navigate to the target, and most importantly, succeed in a new unseen environment. We note that generalization to unseen environments was rarely studied in previous works.

To this end, we quantify the effect of various levels of augmentations, all facilitated by House3D by the means of domain randomization, multi-target training and the diversity of the environment. We resort to well established RL techniques equipped with gating to encode the task at hand. The final performance on unseen environments is much higher than baseline methods by over 8%. We hope House3D as well as our training techniques can benefit the whole RL community for building generalizable agents.

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

| | $|\mathcal{E}|$ | avg. #targets | kitchen% | dining room % | living room% | bedroom% | bathroom% |
|---|---|---|---|---|---|---|---|
| $\mathcal{E}_{small}$ | 20 | 3.9 | 0.95 | 0.60 | 0.60 | 0.95 | 0.80 |
| $\mathcal{E}_{large}$ | 200 | 3.7 | 1.00 | 0.35 | 0.63 | 0.94 | 0.80 |
| $\mathcal{E}_{test}$ | 50 | 3.7 | 1.00 | 0.48 | 0.58 | 0.94 | 0.70 |

Table 2: Statistics of the selected environment sets for RoomNav. `RoomType`% denotes the percentage of houses containing at least one target room of type `RoomType`.

| | test succ. | kitchen% | dining room % | living room% | bedroom% | bathroom% |
|---|---|---|---|---|---|---|
| gated-LSTM | 35.8 | 37.9 | 50.4 | 48.0 | 33.5 | 21.2 |
| gated-CNN | 29.7 | 31.6 | 42.5 | 54.3 | 27.6 | 17.4 |

Table 3: Detailed test success rates for gated-CNN model and gated-LSTM model with "Mask+Depth" as input signal across different instruction concepts.

# A ROOMNAV TASK DETAILS

## A.1 STATISTICS OF SELECTED HOUSE SETS

We show the statistics of the selected three set of houses in Table 2.

In addition to these 5 houses, we also pick another 15 object concepts in our mid-level generalization experiment as auxiliary targets. The object concepts are: *shower*, *sofa*, *toilet*, *bed*, *plant*, *television*, *table-and-chair*, *chair*, *table*, *kitchen-set*, *bathtub*, *vehicle*, *pool*, *kitchen-cabinet*, *curtain*.

**Detailed Specifications:** The location information of an agent can be represented by 4 real numbers: the 3D location $(x, y, z)$ and the rotation degree $\rho$ of its first person view sensor, which indicates the front direction of the agent. Note that in RoomNav, the agent is not allowed to change its height $z$, hence the overall degree of freedom is 3.

An action can be in the form of a triple $a = (\delta_x, \delta_y, \delta_\rho)$. After taking the action $a$, the agent will move to a new 3D location $(x + \delta_x, y + \delta_y, z)$ with a new rotation $\rho + \delta_\rho$. The physics in House3D will detect collisions with objects under action $a$ and in RoomNav, the agent will remain still in case of a collision. We also restrict the velocity of the agent such that $|\delta_x|, |\delta_y| \le 0.5$ and $|\delta_\rho| \le 30$ to ensure a smooth movement.

**Continuous Action:** A continuous action $a$ consists of two parts $a = [m, r]$ where $m = (m_1, \dots, m_4)$ is for movement and $r = (r_1, r_2)$ is for rotation. Since the velocity of the agent should be bounded, we require $m, r$ to be a valid probability distribution. Suppose the original location of robot is $(x, y, z)$ and the angle of camera is $\rho$, then after executing $a$, the new 3D location will be $(x + (m_1 - m_2) * 0.5, y + (m_3 - m_4) * 0.5, z)$ and the new angle is $\rho + (r_1 - r_2) * 30$.

**Discrete Action:** We define 12 different action triples in the form of $a_i = (\delta_x, \delta_y, \delta_\rho)$ satisfying the velocity constraints. There are 8 actions for movement: left, forward, right with two scales and two diagonal directions; and 4 actions for rotation: clockwise and counter-clockwise with two scales. In the discrete action setting, we do not allow the agent to move and rotate simultaneously.

**Reward Details:** In addition to the reward shaping of difference of shortest distances, we have the following rewards. When hitting an obstacle, the agent receives a penalty of 0.3. In the case of success, the winning reward is +10. In order to encourage exploration (or to prevent eternal rotation), we add a time penalty of 0.1 to the agent for each time step outside the target room. Note that since we restrict the velocity of the agent, the difference of shortest path after an action will be no more than $0.5 \times \sqrt{2} \approx 0.7$.

# B EXPERIMENT DETAILS

## B.1 NETWORK ARCHITECTURES

We apply a batch normalization layer after each layer in the CNN module. The activation function used is ReLU. The embedding dimension of concept instruction is 25.

**Gated-CNN:** In the CNN part, we have 4 convolution layers of 64, 64, 128, 128 channels perspective and with kernel size 5 and stride 2, as well as a fully-connected layer of 512 units. We use a linear layer to transform the concept embedding to a 512-dimension vector for gated fusion. The MLP for policy has two hidden layers of 128 and 64 units, and the MLP for Q-function has a single hidden layer of 64 units.

**Gated-LSTM:** In the CNN module, we have 4 convolution layers of 64, 64, 128, 128 channels each and with kernel size 5 and stride 2, as well as a fully-connected layer of 256 units. We use a linear layer to convert the concept embedding to a 256-dimension vector. The LSTM module has 256 hidden dimensions. The MLP module for policy contains two layers of 128 and 64 hidden units, and the MLP for value function has two hidden layers of 64 and 32 units.

## B.2 TRAINING PARAMETERS

We normalize each channel of the input frame to $[0, 1]$ before feeding it into the neural network. Each of the training procedures includes a weight decay of $10^{-5}$ and a discounted factor $\gamma = 0.95$.

**DDPG:** We stack $k = 5$ recent frames and use learning rate $10^4$ with batch size 128. We choose $\alpha_{DDPG} = 100$ for all the settings except for the case with input signal of "RGB+Depth" on $\mathcal{E}_{large}$, where we choose $\alpha_{DDPG} = 10$. We use an entropy bonus term with coefficient 0.001 on $\mathcal{E}_{small}$ and 0.01 on $\mathcal{E}_{large}$. We use exponential average to update the target network with rate 0.001. A training update is performed every 10 time steps. The replay buffer size is $7 \times 10^5$. We run training for 80000 episodes in all. We use a linear exploration strategy in the first 30000 episodes.

**A3C:** We clip the reward to the range $[-1, 1]$ and use a learning rate $1e - 3$ with batch size 64. We launch 120 processes on $\mathcal{E}_{small}$ and 200 on $\mathcal{E}_{large}$. During training we estimate the discounted accumulative rewards and back-propagate through time for every 30 time steps unrolled. We perform a gradient clipping of 1.0 and decay the learning rate by a factor of 1.5 when the difference of KL-divergence becomes larger than 0.01. For training on $\mathcal{E}_{small}$, we use a entropy bonus term with coefficient 0.1; while on $\mathcal{E}_{large}$, the coefficient is 0.05. $\alpha_{A3C}$ is 1.0. We perform $10^5$ training updates and keep the best model with the highest training success rate.

## B.3 GENERALIZATION OVER DIFFERENT CONCEPTS

We illustrate in Table 3 the detailed test success rates of our models trained on $\mathcal{E}_{train}$ with respect to each of the 5 concepts. Note that both models have similar behaviour across concepts. In particular, "dining room" and "living room" are the easiest while "bathroom" is the hardest. We suspect that this is because dining room and living room are often with large room space and have the best connectivity to other places. By contrast, bathroom is often very small and harder to find in big houses.

Lastly, we also experiment with adding auxiliary tasks of predicting the current room type during training. We found this does not help the training performance nor the test performance. We believe it is because our reward shaping has already provided strong supervision signals.

## B.4 AVERAGE STEPS TOWARDS SUCCESS

We also measure the number of steps required for an agent in RoomNav. For all the successful episodes, we evaluate the averaged number of steps towards the final target. The numbers are shown in Table 4. A random agent can only succeed when it's initially spawned very close to the target, and therefore have very small number of steps towards target. Our trained agents, on the other hand, can explore in the environment and reach the target after resonable number of steps. Generally, our DDPG models takes fewer steps than our A3C models thanks to their continuous action space. But in all the settings, the number of steps required for a success is still far less than 100, namely the horizon length.

| | random | concat-LSTM | gated-LSTM | concat-CNN | gated-CNN |
|---|---|---|---|---|---|
| Avg. #steps towards targets on $\mathcal{E}_{small}$ with different input signals | | | | | |
| RGB+Depth (train) | 14.2 | 35.9 | 41.0 | 31.7 | 33.8 |
| RGB+Depth (test) | 13.3 | 27.1 | 29.8 | 26.1 | 25.3 |
| Mask+Depth (train) | 14.2 | 38.4 | 40.9 | 34.9 | 36.6 |
| Mask+Depth (test) | 13.3 | 31.9 | 34.3 | 26.2 | 30.4 |
| Avg. #steps towards targets on $\mathcal{E}_{large}$ with different input signals | | | | | |
| RGB+Depth (train) | 16.0 | 36.4 | 35.6 | 31.0 | 32.4 |
| RGB+Depth (test) | 13.3 | 34.0 | 33.8 | 24.4 | 25.7 |
| Mask+Depth (train) | 16.0 | 40.1 | 38.8 | 34.6 | 36.2 |
| Mask+Depth (test) | 13.3 | 34.8 | 34.3 | 30.6 | 30.9 |

Table 4: Averaged number of steps towards the target in all success trials for all the evaluated models with various input signals and different environments.

