# OpenReview forum: "Building Generalizable Agents with a Realistic and Rich 3D Environment"
_ICLR.cc/2018/Conference — Invite to Workshop Track_

### Official Review · AnonReviewer1 · 2017-11-21
**Not much novelty; overselling the framework and the task**

**Rating:** 4
**Confidence:** 5

**Review:**

Paper Summary: The paper proposes a simulator for the SUNCG dataset to perform rendering and collision detection. The paper also extends A3C and DDPG (reinforcement learning methods) by augmenting them with gated attention. These methods are applied for the task of navigation.

Paper Strengths:
- It is interesting that the paper shows generalization to unseen scenes unlike many other navigation methods.
- The renderer/simulator for SUNCG is useful.

Paper Weaknesses:
The paper has the following issues: (1) It oversells the task/framework. The proposed task/framework is not different from what others have done. (2) There is not much novelty in the paper. The SUNCG dataset already exists. Adding a renderer to that is not a big deal. There is not much novelty in the method either. The paper proposes to use gated attention, which is not novel and it does not help much according to Figures 3b and 4b. (3) Other frameworks have more functionalities than the proposed framework. For example, other frameworks have physics or object interaction while this framework is only useful for navigation. (4) The paper keeps mentioning "Instructions". This implies that the method/framework handles natural language, while this is not the case. This is over-selling as well.

Questions and comments:

- Statements like "On the contrary, we focus on building a flexible platform that intersects with multiple research directions in an efficient manner allowing users to customize the rules and level of complexity to their needs." are just over-selling. This environment is not very different from existing platforms.

- What is referred to as “physics” is basically collision detection. It is again over-selling the environment. Other environments model real physics.

- It is not clear what customizable mean in Table 1. I searched through the paper, but did not find any definition for that. All of the mentioned frameworks are customizable.

- "we compute the approximate shortest distance from the target room to each location in the house" --> This assumption is somewhat unrealistic since agents in the real world do not have access to such information.

- Instruction is an overloaded word for "go to a RoomType". The paper tries to present the tasks/framework as general tasks/framework while they are not.

- In GATED-LSTM, h_t is a function of I. Why is I concatenated again with h_t?

- Success rate is not enough for evaluation. The number of steps should be reported as well.

- The paper should include citations to SceneNet and SceneNet RGBD.

---

> ### Author Response · Authors · 2018-01-04
> **Answer to your concerns**
>
> Dear Reviewer,
>
> We thank you for your feedback. Below we address your comments and point to the changes made to accommodate them.
>
> See general comments (the top official thread) for the novelty of the framework and the definition of the task and instructions.
>
> Unrealistic shortest path:
> 1.     The shortest path is computed only during training to provide the agent with intermediate reward signals. In evaluation, there is no need to compute shortest path (or other statistics involving ground truth) and thus the trained agent can be evaluated in the real world.
> 2.     Even if an agent needs to be trained in the real environment, we can also find surrogate shortest path, e.g., by building a map first. Note that building a map is not a laborious task, since it can be reused many times during training. Again in evaluation, these quantities are not needed.
>
> Physics:
> Currently we only have collision detection and in the future we will add more realistic physics engine (e.g., bullet).
>
> To better reflect the nature of interactions used for RoomNav, we describe the interaction rules in House3D (see Environment Section). However, note that House3D is able to support more complicated engines. For the purpose of our work, we adopt the most lightweight interaction rules for fast experimental cycles. We have updated the changes regarding to physics in the new version of the paper.
>
> Reward shaping
> See general comments.
>
> Analysis of number of steps:
> We have added the analysis for the number of steps. See Appendix B.4. All reported success rates are computed from a fixed number of 100 steps.
>
> Gated LSTM:
> We concatenate the concept I with h_t so that the LSTM module can have direct access to the target when computing its states. In practice, this affected performance by a small amount but it made training more stable.
>
> Missing citations:
> We have cited the relevant environments to our work. See Related Work Section.
>
> To better reflect the motivation and contributions of our work, we have rephrased the parts of the text that seem to cause confusions with the hope that they would clarify hopefully all of your questions.
>
> We sincerely hope that you could read the modified version.

---

### Official Review · AnonReviewer3 · 2017-11-27
**The authors propose a virtual environment built from SUNCG with interactive indoor scenes where agents can perform a variety of human-like tasks. They later focus on the task of navigation, where agents are asked to navigate to a destination without colliding with objects given a high level natural language instruction.**

**Rating:** 5
**Confidence:** 4

**Review:**

Building rich 3D environments where to run simulations is a very interesting area of research.

Strengths:
1.	The authors propose a virtual environment of indoor scenes having a much larger scale compared to similar interactive environments and access to multiple visual modalities. They also show how the number of available scenes greatly impacts generalization in navigation based tasks.
2.	The authors provide a thorough analysis on the contribution of different feature types (Mask, Depth, RGB) towards the success rate of the goal task. The improvements and generalization brought by the segmentation and depth masks give interesting insights towards building new navigation paradigms for real-world robotics.

Weaknesses:
1.	The authors claim that the proposed environment allows for multiple applications and interactions, however from the description in section 3, the capacities of the simulator beyond navigation are unclear.
The dataset proposed, Home3D, adds a number of functionalities over the SUNCG dataset. The SUNCG dataset provides a large number of 3D scanned houses.  The most important contributions with respect to SUNCG are:
- An efficient renderer: an important aspect.
- Introducing physics: this is very interesting, unfortunately the contribution here is very small. Although I am sure the authors are planing to move beyond the current state of their implementation, the only physical constraint currently implemented is an occupancy rule and collision detection. This is not technically challenging.
Therefore, the added novelty with respect to SUNCG is very limited.
2.	The paper presents the proposed task as navigation from high level task description, but given that the instructions are fixed for a given target, there are only 5 possible instructions which are encoded as one-hot vectors. Given this setting, it is unclear the need for a gated attention mechanism. While this limited setting allows for a clear generalization analysis, it would have been good to study a setting with more complex instructions, allowing to evaluate instructions not seen during training.
3.	While the authors make a good point showing generalization towards unseen scenes, it would have been good to also show generalization towards real scenarios, demonstrating the realistic nature of House3D and the advantages of using non-RGB features.
4.	It would have been good to report an analysis on the number of steps performed by the agent before reaching its goal on the success cases. It seems to me that the continuous policy would be justified in this setting.
Comments
-	It is unclear to me how the reward shaping addition helps generalize to unseen houses at test time, as suggested by the authors.
-	I miss a reference to (https://arxiv.org/pdf/1609.05143.pdf) beyond the AI-THOR environment, given that they also approach target driven navigation using an actor-critic approach.


The paper proposes a new realistic indoor virtual environment, having a much larger number of scenes than similar environments. From the experiments shown, it seems that the scale increase, together with the availability of features such as Segmentation and Depth improve generalization in navigation tasks, which makes it a promising framework for future work on this direction. However, the task proposed seems too simple considering the power of this environment, and the models used to solve the task don’t seem to bring  relevant novelties from previous approaches. (https://arxiv.org/pdf/1706.07230.pdf)

---

> ### Author Response · Authors · 2018-01-04
> **Answer to your concerns**
>
> Dear Reviewer,
>
> We thank you for your feedback. Below we address your comments and point to the changes made to accommodate them.
>
> Please check general comments (the very first official thread on the top) for Point 1-2.
>
> 3.  Transfer learning from House3D to real environments (e.g., within an actual building) is an important direction to pursue. Our results also show that using semantic segmentation, depth and RGB images as input, it is possible to have a navigation agent that can be transferred to unseen scenarios. This suggests that using semantic segmentation given by state-of-the-art vision approaches should achieve strong performance without training from raw pixels. As a future work, this could be a promising direction for transfer learning.
>
> 4. We have added the analysis for the number of steps in the Appendix Section B.4. In general, we notice that continuous policies involve fewer steps, as expected.
>
> 5. Stated in the general comments.
>
> 6. There are a few key differences between our task and the task proposed by Al-Thor:
>
> (1)	In AI-Thor, the learned agent is evaluated on the same environments as training, while ours is evaluated on unseen environments. We emphasize that this is a huge difference.
> (2)	In AI-Thor, navigation is restricted within a single room, while our work shows navigation results in houses with multiple rooms and indoor/outdoor situations.
> (3)	AI-Thor designs different networks for different room types, and the target is provided with an actual observation of the object. In contrast, ours use a shared network for 200 houses, and the target is provided with a word (concept). Therefore, the agent needs to associate the concept with the observations.
>
> In summary, our navigation setting poses a major improvement over the setting in AI-Thor, and requires sophisticated actions to achieve the goal. As the first task proposed in House3D dataset, it is not simple at all.
>
> In terms of writing, we have updated the paper and the terminology to clarify the motivation and contributions of our work. Our changes should better reflect the impact of our proposed environment and task.

---

### Official Review · AnonReviewer2 · 2017-11-27
**The paper in general is well written, and the environment will be a useful addition to the community**

**Rating:** 8
**Confidence:** 4

**Review:**

The paper introduces House3D, a virtual 3D environment consisting of in-door scenes with a diverse set of scene types, layouts and objects. This was originally adapted from the SUNCG dataset, enhanced with the addition of a physics model and an API for interacting with the environment.  They then focus on a single high-level instruction following task where an agent is randomly assigned at a location in the house and is asked to navigate to a destination described by a high-level concept (“kitchen”) without colliding with objects.  They propose two models with gated-attention architecture for solving this task, a gated-CNN and a gated-LSTM. Whilst the novelty of the two models is questionable (they are adaptations of existing models to the task),  they are a useful addition to enable a benchmark on the task.  The paper in general is well written, and the environment will be a useful addition to the community.

General Comments
- In the related work section the first part talks about several existing environments.  Whilst the table is useful, for the “large-scale” and “fast-speed”columns, it would be better if there were some numbers attached - e.g.  are these orders of magnitude differences?  Are these amenable to Bayesian optimisation?
- I  didn’t  see  any  mention  of  a  pre-specified  validation  set  or  pre-defined cross-validation sets.  This would surely be essential for hyperparameter tuning
- For the discrete action space state what the 12 actions are.
- The  reward  function  should  be  described  in  more  detail  (can  be  in  appendix).  How is the shortest distance calculated?  As a general comment it seems that this is a very strong (and unrealistic) reward signal, particularly for generalisation.
- There are a number of hyperparameters (αDDPG, αA3C,  entropy bonus terms, learning rates etc).  Some discussion of how these were chosen and the sensitivity to these parameters were helpful
- Figures 3 and 4 are hard to compare, as they are separated by a page, and the y-axes are not shared.
- The additional 3D scenes datasets mentioned by Ankur Handa should be cited.

Typographical Issues
- Page 1:  intelligence→intelligent
- Page  4:  On  average,  there  is→On  average,  there  are;  we  write→we wrote
- Page 7:  softmax-gumbel trick→softmax-Gumbel trick; gumbel-softmax→Gumbel-softmax
- References.  The references should have capitalisation where appropriate.For example,  openai→OpenAI, gumbel→Gumbel,  malmo→Malmo

---

> ### Author Response · Authors · 2018-01-04
> **We sincerely thank you for your comments**
>
> Dear Reviewer,
>
> We thank you for your valuable comments. Note that we have improved the text to better reflect our contributions (see Introduction). We sincerely hope that our proposed House3D environment and RoomNav task will be adopted as a benchmark in RL. We strongly agree that our proposed methods to tackle RoomNav are useful and necessary in order to encourage further research in House3D and concretely describe their extensions from existing RL techniques (see Introduction & Method Section)
>
> We address your comments:
>
> 1. Regarding to simulation speed, our environment is around 1.8K FPS on 120x90 image using only a single M40 GPUs. This makes our environment suitable for RL approaches that typically require both fast simulation and realistic scenarios. To our knowledge, very few environments strike such a balance (e.g., atari games can achieve 6K FPS but its observation is simple, while DeepMind lab and Malmo render more complex images but with a few hundred frames per second, slower than House3D).
>
> 2.     In the first part of our experiments, following standard practice in RL, we trained on 200 house environments and report the success rate on these environments accordingly. Unlike traditional supervised learning, there is no pre-specified validation set or pre-defined cross-validation sets, since every image perceived from the agent in the same house environment can be different.
>
> In the second part of our experiments, we tested our trained agent on 50 unseen house environments, a practice known as transfer learning in RL. We show that the trained agent is able to navigate around unseen environments, much better than other baselines (e.g., random exploration).
>
> 3.      We test both continuous and discrete action space. For discrete action space, there are 8 movement actions and 4 rotation actions with different scales Please see Appendix A for more details.
>
> 4. Sparse rewards pose difficulty in learning, as noted by several RL works in environments that are even simpler than House3D (Mirowski et al., 2016, Jaderberg et al., 2016). As a standard practice in RL, in all our experiments, to guide the agent towards finishing the task, intermediate reward is provided when the agent moves closer to the target, computed by shortest path. Note that this technique, known as reward shaping, is used during training but never during evaluation, in which an agent needs to navigate to the destination alone. Therefore, it does not affect the generalization capability, no matter how strong such a signal is. More details are provided in the Appendix.
>
> 5. The hyper-parameters in our models are tuned using a very rough grid search without extensive tuning.
>
> 6. We have fixed the typos, cited the existing 3D environments mentioned by other commenters and have fixed the figure layout for better comprehension.
>
> We sincerely appreciate your valuable suggestions!

---

### Public Comment · ~Ankur_Handa1 · 2017-10-28
**3D Scenes**

You might also be interested in the following work

- SceneNet https://robotvault.bitbucket.io/
- SceneNet RGB-D https://robotvault.bitbucket.io/scenenet-rgbd.html

They are photorealistic, 3D, customisable and potentially large scale. The source code is here https://bitbucket.org/dysonroboticslab/scenenetrgb-d/src. I think these papers are worth citing too.

Matterport3D is a recent paper which has real 3D scenes https://niessner.github.io/Matterport/. This is also worth citing.

You could talk about the limitations/benefits of these datasets in your table.

---

> ### Author Response · Authors · 2017-11-05
> **Really appreciate your suggestion**
>
> Thank you very much for bringing these datasets to our attention. They are indeed relevant and very valuable to our work. We promise to cite all of them in the next version.
>
> Actually, the design of our environment is flexible and can utilize 3D models from different sources. We have done some initial work to incorporate Matterport3D to our environment and our goal is to release a general API that can be used to incorporate a variety of 3D model sources, including SceneNet.

---

### Public Comment · ~Ankesh_Anand1 · 2017-11-03
**Open-sourcing the environment?**

Very neat work. We would be very interested in working with your environment. Would you be able to publish the source code / dataset for the work anonymously using a service like http://anonymous.4open.science/ ?

---

> ### Author Response · Authors · 2017-11-05
> **We will open-source it ASAP**
>
> We will provide flexible APIs as soon as possible, once the codebase of the environment is cleaned up.

---

> ### Public Comment · ~Saurabh_Gupta1 · 2017-11-23
> **Alternative Environments**
>
> Hi Ankesh,
>
> In the meantime, you could try out our realtime 3D environment simulation and rendering code (https://github.com/tensorflow/models/tree/master/research/cognitive_mapping_and_planning).
>
> For our CVPR 17 paper (https://sites.google.com/view/cognitive-mapping-and-planning/), we used it for visual navigation tasks using Matterport Scans from the Stanford Building Parser Dataset (http://buildingparser.stanford.edu/index.html), but it should be easy to plug in other sources of mesh data.

---

### Public Comment · ~Ankesh_Anand1 · 2017-11-25
**Questions regarding the environment**

Hi,
I had a few questions regarding the environment:

- Interaction: What agent-object interactions does House3D implement? Is there interaction aside from collision detection?
-Physics Engine:  Is there support for rigid body dynamics, and can objects react to external forces and gravity?

---

### Author Response · Authors · 2018-01-04
**General Clarifications to Common Misunderstandings**

Dear Reviewers and AC
We note some common misunderstandings in the reviews, so we highlight and clarify our main contributions in this thread. We also improved our introduction section with all these points clarified.

**Novelty and Contributions**
 In addition to novel algorithms, we believe that it is also important to acknowledge the contributions regarding to environments and implementations. These contributions also help tremendously to research community, not by solving technically challenging problems, but by pointing to the right directions to pursue. For example, ImageNet fuels the usage of DL models in computer vision research because of its scale; AlphaGo achieves super-human Go AI by combining many old ideas in RL (e.g., ConvNet, MCTS, Selfplay) and massive computational resource together. An algorithm-centric criterion of novelty would reject both of the great works, and might lock researchers in the circle of “smart algorithms”that might look mathematically interesting but never generalize well in the case of large scale and complicated situations.

 A main contribution of our work is that we are the first ones to explore “semantic-level” generalization in RL. Semantic-level generalization studies an agent’s ability to extract conceptual abstractions (e.g., kitchen) from observations from a diverse set of scenes, and apply the same abstraction in unseen environments. Note that this is contrary to popular definitions of generalization in the RL literature such as pixel-level perturbations (e.g., object colors) or levels of difficulties (e.g., maze configurations).

 To explore semantic-level generalization, we develop a suitable large-scale environment, House3D. We focus on its scale (45k human-labored houses from SUNCG), its flexibility and efficiency. House3D can render images of 120x90 resolution on a single M40 GPU with 1600fps. As a testament to its flexible design, House3D has already been used for tasks beyond navigation, such as embodied QA (Das et al., 2017a). Moreover, our platform is not restricted to SUNCG, but can also use other data sources (e.g., Matterport, SceneNet).

 As an attempt to study semantic-level generalization in House3D, we define a “concept-driven navigation” task, RoomNav. Here the goal is conveyed by a concept rather than a natural language instruction. We propose RoomNav in order to evaluate whether an agent can understand “semantic concepts” (e.g., room types) and can generalize to unseen scenarios. We hope RoomNav can serve as a benchmark task for semantic generalization.

 To tackle RoomNav, we propose to use gated-attention networks, which are shown to be effective for this task and can potentially serve as strong baselines for further benchmarking and fair comparisons.

**Concept/Instructions**
In the submitted version of this paper, we refer“instruction”as the concept (e.g., roomtypes) that the agent needs to associate with its observations during exploration. We emphasize that in the paper, we did not suggest any connections between instruction and natural language. In fact, natural language instructions are beyond the scope of this submission. We have updated Method Section (Sec. 4) for a clearer narration to avoid possible confusion (as R2/R3’s comments suggest).

**Reward Shaping**
Reward shaping, as we state in the paper, provides a supervisory signal that helps the agent to learn faster. It is not provided to the agent during test time. Note that the RoomNav task is difficult and a sparse reward hinders training. We actually tried several reward-shaping approaches but didn’t see any impact on generalization in experiments. So we only report the most effective reward shaping in the paper.

We have updated the abstract, introduction and related work of the paper to clarify these points.

---

### Decision · Program_Chairs · 2018-01-29
**ICLR 2018 Conference Acceptance Decision**

**Decision:**

Invite to Workshop Track

**Comment:**

The authors present an environment for semantic navigation that is based on an existing dataset, SUNCG. Datasets/environments are important for deep RL research, and the contribution of this paper is welcome. However, this paper does not offer enough novelty in terms of approach/method and its claims are somewhat misleading, so it would probably be a better fit to publish it at a workshop.